# Durability of the Exterior Transparent Coatings on Nano-Photostabilized English Oak Wood and Possibility of Its Prediction before Artificial Accelerated Weathering

**DOI:** 10.3390/nano9111568

**Published:** 2019-11-05

**Authors:** Miloš Pánek, Štěpán Hýsek, Ondřej Dvořák, Aleš Zeidler, Eliška Oberhofnerová, Kristýna Šimůnková, Přemysl Šedivka

**Affiliations:** Department of Wood Processing and Biomaterials, Faculty of Forestry and Wood Sciences, Czech University of Life Sciences in Prague, Kamýcká 129, 165 00 Prague, Czech Republic; panekmilos@fld.czu.cz (M.P.); hyseks@fld.czu.cz (Š.H.); dvorak18@fld.czu.cz (O.D.); zeidler@fld.czu.cz (A.Z.); Eliska.Oberhofnerova@seznam.cz (E.O.); simunkovak@fld.czu.cz (K.Š.)

**Keywords:** oak wood, surface modification, UV-stabilization, nanoparticles, contact angle, surface free energy, exterior transparent coatings, durability

## Abstract

Changes in surface material characteristics can significantly affect the adhesion and overall life of coatings on wood. In order to increase the durability of transparent exterior coatings, it is possible to use the surface modification of wood with UV-stabilizing substances. In this work, selected types of surface modifications using benzotriazoles, HALS, ZnO and TiO_2_ nanoparticles, and their combinations were applied to oak wood (*Quercus robur*, L.). On such modified surfaces, the surface free energy, roughness, and contact wetting angle with three selected types of exterior transparent coatings were subsequently determined. An oil-based coating, waterborne acrylic thick layer coating, and thin-layer synthetic coating were tested and interaction with the aforementioned surface modifications was investigated after 6 weeks of accelerated artificial weathering. The results of changes in the initially measured surface characteristics of the modified oak wood were compared to the real results of degradation of coatings after artificial accelerated weathering. The positive effect of surface modification, in particular the mixture of benzotriazoles, HALS, and ZnO nanoparticles on all kinds of coatings was proven, and the best results were observed in thick-film waterborne acrylic coating. The changes in the measured surface characteristics corresponded to the observed durability of the coatings only when measured by wetting using drops of the tested coatings.

## 1. Introduction

Changes in the surface characteristics of wood can significantly affect the adhesion of coatings and their overall durability during exterior weathering [1,2,3]. The adhesion of coatings to wood tends to be significantly influenced by the type of underlying wood species [2], its moisture content [4], the roughness and the processing method [5], the polymeric base of the coatings and the added additives [6], but also by the application quality. The long durability of the coating system is a necessary condition for protecting wood against abiotic weathering [7], but also against biotic damages by bacteria, microscopic fungi, wood-destroying fungi, and other bio-degrading agents [8,9]. Modern exterior, highly-pigmented coating systems are characterized by a relatively long durability [10]. However, transparent coatings on wood have not yet been successful in addressing long-term durability during outdoor exposure fully exposed to precipitation, sunlight, and other degradation effects [11]. Saving the native wood’s appearance could also be very interesting due to its positive effect on the human psyche [12]. There are several solution investigation paths for this purpose—(a) surface modification of the underlying wood using nanoparticles, UV-stabilizers, HALS (hindered amins light stabilizers), or fungicides [11,13,14]; (b) modification of paints themselves with suitable additives—UV-stabilizers, HALS, nanoparticles, transparent pigments and others [15,16,17]; (c) appropriate coating system layering technology [18,19] and the use of top hydrophobic layers [20]. Other research methods use the growing of ZnO or TiO_2_ nanoparticle films on wood surfaces [21,22,23] or the creation of photo matrix constituents [24]. More work has been devoted to the determination of wood surface characteristics, the contact angle of wetting, and surface free energy, all of which affect adsorption and thus the adhesion of coatings to wood [25,26,27,28]. These surface characteristics are changed by the surface modification of the underlying wood species, but also by its aging during exposition [4,29,30,31]. In both cases, the effect of changes in the chemical composition of the substrate is visible [32,33,34,35]; with regard to modifications with nanoparticles, this consists of a change in wetting due to nanosized structural changes in surface morphology [36]. The adhesion of wood transparent coatings during exterior exposure and their total durability is strongly influenced by the decomposition of lignin and extracts due to the UV and visible (VIS) spectrum of sunlight penetrating these types of coatings [11,33,34]. Based on previous research [37], this research investigated the possibility of surface modification of the underlying type of wood (oak—*Quercus robur*, L.) and mainly interaction with the applied top protective transparent exterior coating. UV-stabilizers benzotriazols, HALS, ZnO and TiO_2_ nanoparticles, and their combinations with tested effects on slowing degradation under the influence of UV radiation were used as the first penetrating layer [13,23,37,38]. In addition, oak wood is characterized as durable against bio-damages, but on the other hand by a problematic reaction to coatings due to a complicated morphological structure with large open vessels [39] and a relatively high tannin content affecting the longevity of exterior coatings [15,40]. The total durability of exterior coatings is greatly influenced by the type of underlying wood [41], but also by the coating itself and its polymer base associated with its penetration [6] and mainly by used additives [7,11]. For its confirmation before use in practice, it is possible to use accelerated weathering tests in UV-chambers or Xenotests, which can also be confirmed by long-term multi-year tests of natural weathering in an exterior [42,43,44,45].

The main aim of this work is to research the interaction between different surface modification increasing the photostability of wood and three types of transparent coating systems. The second aim is to find out whether it is possible to quickly predict the durability of the tested coatings during weathering on the basis of the evaluation of selected surface characteristics of wood/modified wood changes. If this assumption is confirmed, then it would be possible to select unsuitable types of treatments prior to accelerated laboratory and long-term tests using artificial or natural weathering. For this purpose, and for a more thorough analysis, multiple combinations of surface modification enhancing oak wood photostability and their effect on extending the overall durability of three types of exterior transparent coatings on different bases were tested. Selected surface characteristics—any change of roughness, surface free energy of the underlying un/modified wood species, and changes in the contact wetting angle using selected types of coatings were compared to the overall durability of these coatings after accelerated laboratory weathering using a combination of UV-radiation, water spraying, and temperature cycling.

## 2. Material and Methods

### 2.1. Wood Samples

Oak wood samples (*Quercus robur* L.) with dimensions of 40 mm × 20 mm × 150 mm (*T* × *R* × *L*) and mean density ρ_0_ = 708 kg/m^3^ (moisture content of 12%) were used in this experiment. The samples were treated using sandpaper with a grit of 120 in a longitudinal direction, and they were visually sorted in order to minimalize the color variability of the tested wood material. The cross ends of samples were sealed using silicon and surface modifications and the tested coatings were subsequently applied.

### 2.2. Surface Modification and Coatings

Four different surface modifications (in 3% concentration in the form of water solution or dispersion) were applied in one layer in the amount of ≈ 100 g/m^2^ using a brush (Table 1).

These surface modified samples (M1–M4) and reference control samples (Ref—without modification) were subsequently coated with three different transparent commercial paints, which were applied in two layers in the amount of 120 g/m^2^ using a brush according to a recommendation from a manufacturer (Table 2). SEM and confocal laser scanning microscopy have shown that dry film thickness was approximately 30 µm for acrylic and 5 µm for penetrating oil and synthetic coatings. Two samples were tested for each type of surface modification and coating system. The mean values of the initial color of the tested samples and coating systems are given in Table 3.

### 2.3. Artificial Weathering

Artificial weathering was carried out in UV-chamber QUV (Q-Lab, Cleveland, OH, USA) on the basis of modified EN 927-6 [46] (Table 4). During each weekly cycle of irradiation and spraying, the samples were transferred to the conditioning chamber Discovery My DM340 (ACS, Massa Martana, Italy) and exposed to three-hour cycles lasting 6 h in total using temperature changes from −25 to 80 °C (with 25% relative air humidity). This led to better imitation of the exterior conditions in a mild climatic zone, and to acceleration of artificial weathering tests. The total weathering time consisted of 1008 h (6 weeks) of UV-cycling and water spraying, and 36 h of temperature cycling.

### 2.4. Analyses of Tested Wooden Surfaces

The color parameters defined in CIE 1986 [47] of the tested samples were measured after 1, 3, and 6 weeks of weathering using Spectrophotometer CM-600d (Konica Minolta, Osaka, Japan). The device was set to an observation angle of 10°, d/8 geometry and D65 light source, and the SCI method was used. Six measurements per sample exposed to artificial weathering were carried out for each weathering time.

According to the Euclidean distance, the total color difference ∆*E** (CIE 1986) was subsequently calculated using Equation (1):(1)$$\Delta E*=\sqrt{∆L^{*2}+∆a^{*2}+∆b^{*2}}$$
where ∆*L**, ∆*a**, and ∆*b* a* are relative changes in color between the initial and weathered state; *L** is lightness from 0 (black) to 100 (white), *a** is chromaticity coordinate + (red) or − (green), and *b** is chromaticity coordinate + (yellow) or − (blue).

Gloss changes were evaluated using glossmeter MG268-F2 (KSJ, Quanzhou, China) on the basis of EN ISO 2813:2014 [48]. Three measurements at a 60° angle per sample after 1, 3, and 6 weeks of weathering were carried out.

Surface free energy (SFE) and contact angles (CA_coating_°, CA_water_°) was evaluated using goniometer Krüss DSA 30E (Krüss, Hamburg, Germany) with software Krüss (Krüss, Hamburg, Germany) and ORWK model for determination of SFE in mN.m^−2^. The sessile drop method with a dosing volume of liquids 5 µL was used in all cases. Distilled water was used as polar liquid and diiodomethane as non-polar liquid for SFE evaluation. The contact angle measurements with distilled water were done at 5 s after deposition on the basis of other studies [49,50,51]. Due to faster penetration of diiodomethane into the oak wood surfaces, the contact angle of this liquid was measured during the first second after deposition of drop. The contact angle (CA_coating_°) of coating drop at 5 s after the deposition on oak wood surfaces (modified (M1–M4) or reference unmodified (Ref) was also measured in order to compare the wettability of different surface modifications with the tested coatings. The wettability of coated surfaces against water (CA_water_°) indicating changes in hydrophobicity was measured after 0, 1, 3, and 6 weeks of artificial weathering with distilled water. The dynamic water contact angle water was measured during 120 s using 5 µL of distilled water on samples before and after accelerated weathering. Ten measurements per each tested type of sample and all kinds of measurements (SFE, CA_coating_°, CA_water_°) were done.

The roughness parameter *R_a_* of oak wood surfaces after modification was measured using confocal laser scanning microscope and profilometer Lext Ols 4100 (Olympus, Tokyo, Japan) on the basis of EN ISO 4287:1997 [52] and EN ISO 4288:1996 [53]. The measurement was carried out in four traversing lengths oriented perpendicularly to the length of the samples over the tangential surface.

### 2.5. Microscopic and Elemental Composition Analyses, Visual Analyses

The selected sections of the wood-penetration layers and tested coatings were observed with a MIRA 3 electron microscope (Tescan Orsay Holding, Brno, Czech Republic) with a secondary electron detector operated at 15 kV acceleration voltage. The elemental compositions of the tested sections were examined by an energy dispersive spectroscopy system (Bruker XFlash X-ray detector, Karlsruhe, Germany, and ESPRIT 2 software). Hydrogen is not detectable by the method used.

Surfaces were additionally scanned at the beginning and after 1, 3, and 6 weeks of artificial weathering using a Canon 2520 MFP scanner with 300 DPI resolution (Canon, Tokyo, Japan) to evaluate visually and save degradation of the tested coating systems.

### 2.6. Statistical Evaluation

Statistical analyses were evaluated in MS Excel and Statistica (StatSoft, Palo Alto, CA, USA) using mean values, standard deviations, line plots, whisker plots with mean values and 95% two-sided confidence intervals. The results were statistically compared using a Tukey HSD test at a 95% significance level.

## 3. Results and Discussion

### 3.1. Surface Characteristics of Wood after Modification

The first part of the research was focused on the evaluation of changes in the surface characteristics of oak wood after application of modifying solutions M1–M4 (see Materials and Methods) and to compare them with unmodified wood. Effective types of modifications increasing the color stability of wood under the effects of the UV + VIS spectrum were selected on the basis of previous work [37]. The roughness change (Figure 1), surface free energy change (Figure 2), and the change in the contact angle of wetting with the tested coatings (Figure 3) were evaluated.

Wood roughness after application of aqueous solutions with UV stabilizers slightly increased (statistically significant) (Figure 1) compared to untreated wood in two cases (M1 and M3). This is due to the elevation of damaged wood fibers on the surface after increasing the moisture by applying the aqueous solution [54]. The change in roughness between the modifications was almost the same—the differences were statistically insignificant, confirming the above hypothesis (0.89 < *p* < 1.00). Different roughness of wood can alter the adhesion of coatings [2,5,55]. A slight increase in adhesion strength [2] was observed in the aforementioned works for chestnut with an increase in *Ra* from 4.5 to 8.3 µm. In the work of Vitosyté et al. [5], the decrease of *Ra* to ash wood from 8.64 to 4.59 µm led to only a slight increase in adhesion strength. In the work of Jaić et al. [55], the influence of the grinding direction was more significant, although a slight effect of adhesion was observed when *Ra* increased from 3.5 to 4.8 µm. Generally, a slight increase in *Ra* improves adhesion, but a large increase of *Ra* markedly worsens it. According to the cited works [2,5,55], the measured increase in *Ra* after modification M1–M4 (Figure 1) should not negatively affect the adhesion of coatings on painted wood, and thus negatively affect peeling during exposure.

Modifications of the underlying wood can significantly change the SFE values [29,30,50]. The SFE of oak wood increased significantly after all surface modifications (Oak-REF = 40.12 mN.m^−2^ to values from 63.68 mN.m^−2^ to 74.38 mN.m^−2^ for M1–M4). Among the modifications, only M4 (SFE = 74.88 mN.m^−2^) differed significantly from M1 (SFE = 63.68 mN.m^−2^); otherwise there were no statistically significant differences. The increase in SFE predicts better wettability of surfaces with a coating that is associated with better adsorption, and hence adhesion [6,25,26]. The improvement in wettability of the tested coatings (CA_coating_° of Acryl and Oil) corresponded to the increase in SFE after modification, with the exception of M3, where, despite the highest increase in SFE (Figure 2), the contact angle of wetting of the coatings increased even above the unmodified oak wood value (Figure 3). In M3, TiO_2_ nanoparticles with smaller dimensions (6 nm) were used compared to ZnO nanoparticles (≈40 nm) and therefore the effect of worse wettability by coatings, which did not occur in water and diiodomethane during SFE measurements, could occur. Increasing the contact angle of wetting due to the structural arrangement of the surfaces is described in several works [36,56], where the arrangement and the distance of the individual structural units on the surface seems to be crucial. From the perspective of the prediction of adhesion during exposure, an improvement in the coating system properties (especially peeling) of all modifications (M1–M4) can be expected based on the SFE changes (Figure 2). Based on the evaluation of the wettability changes with the tested coatings (Acryl and Oil; while for the Synthetic coating, CA_coating_° could not be measured due to instantaneous infiltration into wood), an increase in adhesion of modifications M1, M2, and M4 can be expected, whereas worsening of M3 is assumed compared to an unmodified surface (Oak-REF) (Figure 3).

These assumptions were subsequently evaluated using accelerated artificial weathering tests in a UV-chamber with embedded thermal cycling (see Materials and Methods). The changes in color (Figure 4), gloss (Figure 5) and contact angle of wetting (Figure 6 and Figure 7) were evaluated, which are the characteristics defining the degree of degradation of the coating systems and also the underlying wood species [57,58,59]. The evaluation was supplemented by an SEM and elemental analysis (Figure 8) of selected coating systems with different results and an overall visual assessment of all tested surfaces during weathering (Figure 9). Visual assessment is required by standards (for example the European standard EN 927-6 [46]) and also provides a complete basis for determining the degree of damage after weathering in research papers [60,61].

### 3.2. Changes in Color and Gloss during AW

Application of transparent coating systems on oak wood causes darkening and increasing of red and yellow shades, mainly in the case of synthetic coating (Synth) and partly oil-based coating (Oil). Some differences were observed when the initial M1–M4 surface modifications were applied (Table 3). The effects of the UV-stabilizing modification (mainly M1) of the surface of the underlying wood were most evident in the increase in the color stability of the acrylic thick-layer glazing, which was maintained even after 6 weeks of AW. The variability of the measured color changes was also lower using surface modifications compared to acrylic coating applied to unmodified oak wood (Figure 4A). For synthetic thin glazing and oil-based coating, the color change was significantly influenced by degradation and peeling of the paint after 3 weeks of AW, and the effect of the underlying modification was not clear (Figure 4B,C). In the reference oak without coating (REF), the color change was more pronounced, especially after 3 and 6 weeks of accelerated weathering, due to the leaching of photodegraded extractives and lignin [33,34]. For the tested coatings, the significant change *ΔE** suggests the same phenomenon after damage to the coating layer, which is also confirmed by the visual assessment of degradation (Figure 9). Overall, the tested surface modifications increased the color stability of the test specimens in more cases [13,14], in particular the M1 wood modification was able to stabilize the underlying wood against color changes associated with the photodegradation of lignin and extractive substances (Figure 4).

Gloss changes indicate that the top surfaces of the coating layer were damaged by weathering [57,59]. The Oil and synthetic coatings were matte as defined in EN ISO 2813:2014 [48] and their low gloss was reduced even after 1 week of AW (Figure 5B,C). This indicates their rapid degradation. The lower layer thickness compared to the acrylic thick-layer glazing (see Figure 8) led to the deterioration of the overall appearance, as confirmed by the visual evaluation (Figure 9). The acrylic (Acryl) also showed a decrease in gloss (Figure 5A). Using the underlying modification (M1–M4), the gloss reduction rate was reduced, but only slightly. After 6 weeks of weathering, the gloss dropped from G ≈ 34 values initially to values from G ≈ 18 to G ≈ 25.

### 3.3. Changes in Water Contact Angle during AW

The change in CA_water_° via water indicates a faster degradation of the thin-layer synthetic glazing (Synth) and the oil coating (Oil) compared to the thick-layer acrylic glazing (Figure 6). A decrease in wettability to 0° indicates total degradation of the coating or surface layers of wood due to weathering [28,62]. A more pronounced decrease from the initial values (CA_water_° = from 100° to 80°) also indicates an impairment of the protective function of the coating against water [59,63]. A sufficiently thick layer of glazing can provide longer-term protection against weathering (Figure 6A). No significant differences were observed in modified (Acryl-Ref) and unmodified wood (Acryl M1–M4). In the oil coating (Oil), the initial M4 modification that most slowed down decreasing of hydrophobicity after 6 weeks of AW (Figure 6B), and for the synthetic glazing, was modification M2 (Figure 6C). Dynamic water contact angle measurements confirm these results (Figure 7). The acrylic coating has a good protective function after 120 seconds also after 6 weeks of weathering and differences between the unmodified and M2, M3 modified surfaces were negligible (Figure 7A). Oil-based and synthetic coatings have better results for unmodified and M2 modified surfaces, where some protective function against water was observed. Modification M3 increases the degradation of coating layer and the water drops soaked into wood surfaces very quickly (Figure 7B,C). For oak, it was confirmed that otherwise suitable and long-lasting transparent oil coatings, proven on other types of wood [64], do not produce sufficiently good results. Overall, based on several works, oak can generally be characterized as a type of wood with a difficult protection finish affected by exterior transparent coatings [15,40,59].

### 3.4. SEM and Visual Analyses

The SEM and elemental analysis confirmed the deposit of nanoparticle surface modifications in oak wood surface layers. It also showed that penetration of treatments and tested coatings was achieved only into the first layer oak wood cells destroyed during sanding (Figure 8). In unmodified oak wood with acrylic coating (Acryl-Ref, Figure 8A), SEM did not confirm any significant damage to the coating layer compared to the more stable color modification (see Figure 4) under the acrylic coating (Acryl-M2, Figure 8B). However, it is possible to see a significant disturbance in the wood-coating system interlayer compared to the M2 modified surface (Figure 8A versus Figure 8B). This was also confirmed via a visual analysis of the entire surface, where more frequent disturbances were visible on the test surface of the sample Acryl-Ref (Figure 9). For comparison, a more degraded synthetic coating (Synth-M3) (Figure 8C) was also evaluated using SEM and elemental analysis. It is apparent that the surface layer was damaged after AW, but no UV-stabilizing nanoparticles (TiO_2_ and ZnO) were washed out in the area where the coating was penetrated, and the hydrophobicity of the surface was partially preserved (Figure 6C). However, the aesthetic functionality and color of the surface was significantly impaired (Figure 4C and Figure 8).

The visual evaluation (Figure 9) gives an overall view of color change, coating flaking, cracking and complete defoliation [58,60]. The significant effect of the applied film-forming substance and the type of coating used was confirmed [7,11]. In particular, the effect of the thickness of the top glazing layer, which decreases during exposure due to weather, was noticeable [43]. The results of lower color change (Figure 4) and CA_water_° changes (Figure 6) for acrylic glazing (Figure 9) were confirmed. The visual evaluation also confirmed the positive effect in particular of M2 and M4, and partially of M1 surface modifications on the overall durability of this coating. The reference on unmodified oak (Acryl-Ref) showed greater crack formation compared to them. The prediction of faster damage of coating based on a higher contact angle of wetting with coating material was confirmed for modification M3 (Figure 3 versus Figure 9). Adversely, this prediction using *R_a_* and SFE changes was not confirmed (Figure 1 and Figure 2 versus Figure 9). Oil and synthetic coatings were shown to have improved durability and color stability by using modifications M2 and partly M1 and M4, in particular after 3 weeks of weathering (Figure 9). After 6 weeks of AW, the effect was observed only by lower degradation of the underlying wood and both coatings (Oil and Synthetic) were completely degraded. The results confirm that in poorly permeable woods, due to weathering, the penetration and thin-film coatings are rapidly degraded and their faster renewal is necessary [20]. The effect is even more pronounced for wood species with a high content of extractives and an uneven morphological structure [40,58,60].

### 3.5. Final Discussion

Of the possible methods of predicting the rate of damage to coating systems on modified wood during exposure, it was confirmed that only surface wetting with specifically applied coatings (CA_coating_°) is diagnostically appropriate (Figure 1, Figure 2 and Figure 3, versus Figure 9). However, the method is limited to coatings that do not immediately penetrate the wood and where the CA_coating_° is measurable. This result is consistent with the work of de Meier and Militz [1]. The measured SFE and *R_a_* values on modified oak wood surfaces did not predict the achieved durability and coating defoliation results during AW (Figure 1 and Figure 2 versus Figure 9). In previous works [1,28], where SFE was evaluated, and in some of them also its effects on coating adhesion, the differences in native wood (in the range from 40 to 55 mN.m^−2^) are too small to significantly affect the adhesion of coatings [6]. The results of this work confirm that not even a statistically significant increase in SFE after surface modification of 50% or more (Figure 2) has to clearly predict the defoliation of coatings during exposition (Figure 9). It must be mentioned that the loss of coating adhesion during weathering is a complex phenomenon related not only to the change in surface characteristics of the underlying wood species and its photodegradation [33,34], but also to depolymerization and loss of internal cohesion of the coating system [11]. This can be significantly influenced by additives [13,16], but they may also have a negative impact on the compactness of the polymer base of the coating [65]. This could also be the reason for infective wood modification in this work using small TiO_2_ nanoparticles (Figure 9), and their specific placement in the coating system structure seems to be necessary [66]. The results also show that modification of the underlying wood is a promising option for increasing the durability of transparent exterior coating systems [13,67]. This is confirmed by the results of the work of Evans et al. [40], where, based on a multifactorial analysis of the effect of (a) thickness, (b) UV-stability and flexibility of the coating, (c) dimensional stability, and d) photo-stability of the underlying wood on the durability of transparent coatings, the use of suitable surface modifications enhancing the photostability of the wood proved to be a key factor. Our work confirmed the positive effect of surface modifications using UV-stabilizing agents even on oak wood with a high content of extractives [68] and strongly inhomogeneous morphological structure [39], which lead to faster degradation of exterior coatings compared to other kinds of underlying wood [15,40]. The tested synthetic thin-film coating had the shortest durability, and the oil also shorter compared to the acrylic thick-layer glazing. The advantage of oil coatings, however, is their easier renewability. In terms of renovation, there is no need for complete grinding, which is necessary for heavily damaged acrylic glazes. The acrylic water-solvent coating system provided good results, and the color stability during AW was further enhanced by the surface modification. A big advantage of this coating is low VOC content and health safety [69,70].

The results of this study show the importance of researching the interaction between effective UV-protective surface modifications/treatments and applied coating systems. The use of nanoparticles with multifactorial effect not only for UV but also for bio-protection of outdoor exposed wooden surfaces [71] is promising. The use of combinations of UV-stabilizing substances for underlying wood or top coating systems seems to be more advantageous than the use of each of them separately. The work of Rao et al. [72] confirmed the use of a combination of ZnO nanoparticles with benzotriazoles, and this presented study also the combination of benzotriazoles, HALS, and ZnO nanoparticles. In this work, the prediction of long-term durability of the coating system for certain type of modifications with UV-stabilizers, some types of coating materials and *Q. robur* wood was confirmed. However, it will be necessary to test and confirm the results for other types of wood, other types of modifications, and other types of coating materials in order to confirm the general validity of the used method.

## 4. Conclusions

Transparent coatings on exterior wood, in particular oak (*Quercus* sp.) have low overall durability. In this work, the positive effect of surface modifications of oak wood was confirmed in particular by the combination of ZnO nanoparticles with benzotriazoles and HALS on the overall durability of the selected coating systems. The importance of researching the interaction between effective UV-protective surface modifications/treatments and applied coating systems was shown. The best results were achieved with an acrylic waterborne thick-layered glaze, followed by an oil-based coating, and the worst variant tested was a thin-layer synthetic glaze. The possibility of predicting the overall durability of the coating system on modified wood using wettability applied by the coating drop was confirmed. Conversely, the effect of the change in the surface free energy of the wood and the change in roughness due to the modification did not correspond to the changes in the overall durability of the tested coating systems. The method of fast initial prediction is useful in research focused on increasing the durability of transparent coating systems using the modification of underlying wood. This enables, at the beginning, the elimination of disadvantageous and ineffective variants from accelerated laboratory, as well as long-term natural weathering testing.

## Figures and Tables

**Figure 1 nanomaterials-09-01568-f001:**
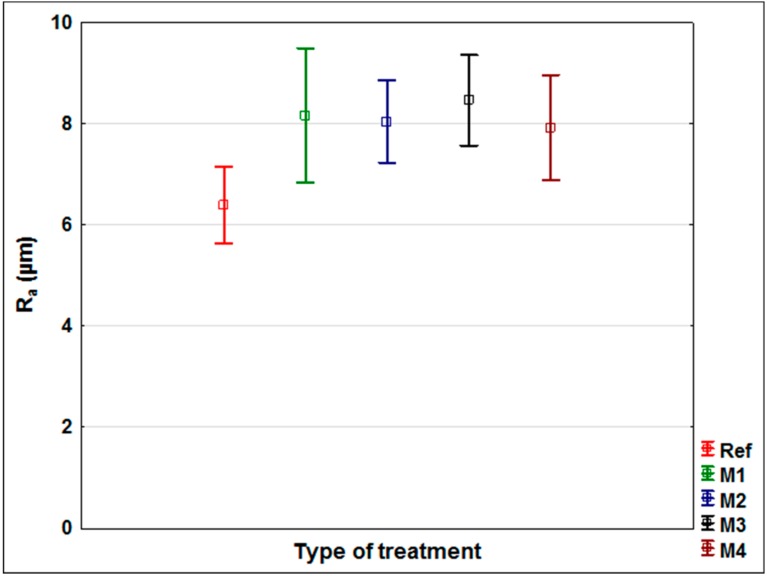
Roughness (R_a_) of oak wood samples without (Ref) and after surface modification (M1–M4). The Tukey HSD test shows that the differences in the analysed values were statistically significant (*p*-value < 0.05) for M1 and M3 compared to Ref.

**Figure 2 nanomaterials-09-01568-f002:**
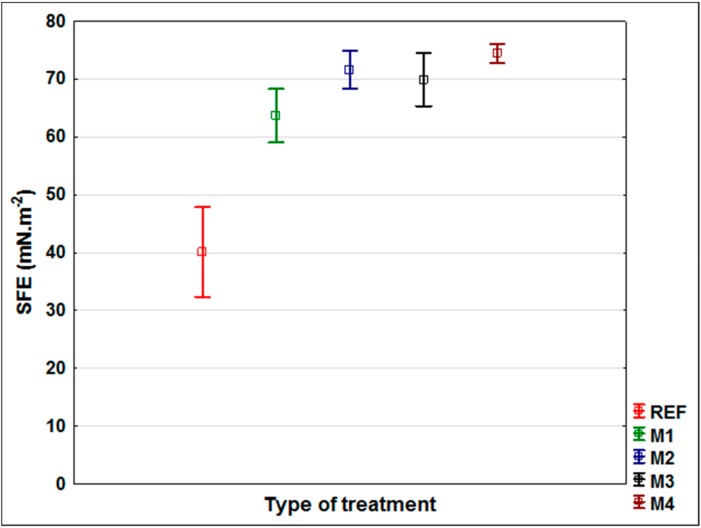
Surface free energy (SFE) of oak wood samples without (Ref) and after surface modification (M1–M4). The Tukey HSD test shows that the differences in the analysed values were statistically significant (*p*-value < 0.05) for all modification (M1–M4) compared to Ref and also between M1 and M4.

**Figure 3 nanomaterials-09-01568-f003:**
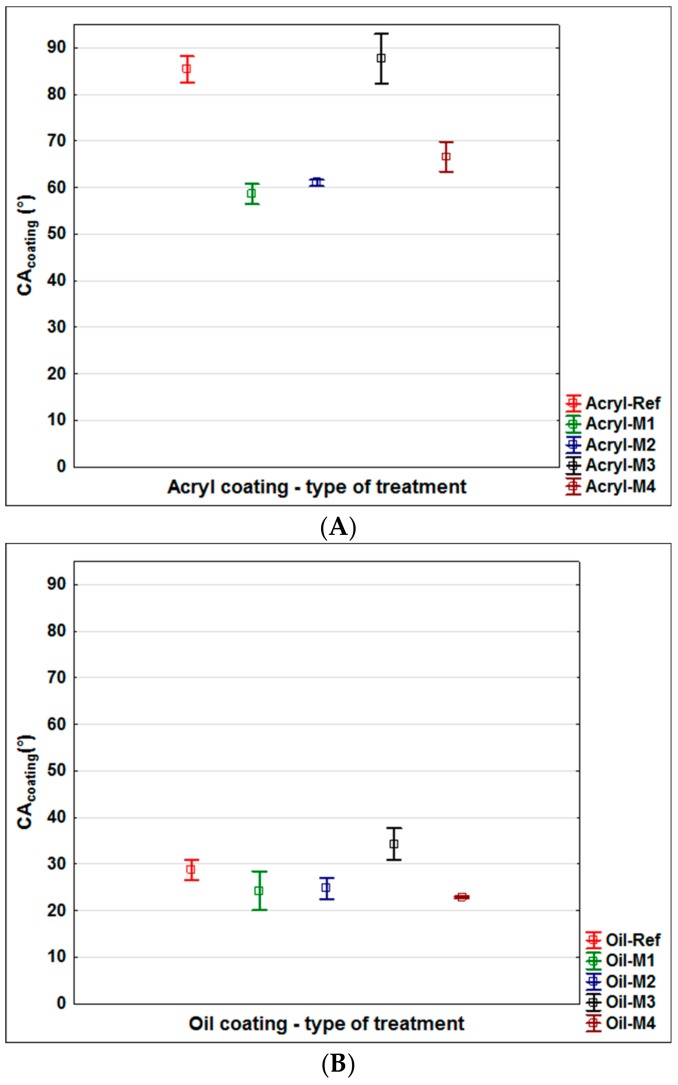
Contact angle (CA_coating_°) of the tested paints on oak wood samples without (Ref) and after surface modification (M1–M4). (**A**): For acrylic paint, the Tukey HSD test shows that the differences in the analysed values were statistically significant (*p*-value < 0.05) for modification (M1, M2, M4) compared to Ref and also between M3 and M1, M2, M4. (**B**): For oil-based paint, the Tukey HSD test shows that the differences in the analysed values were statistically significant (*p*-value < 0.05) for modification (M3 and M4) compared to Ref and also between M3 and M1, M2, M4. Note: The contact angle of Synthetic paint (Synth) could not be measured due to its very fast soaking into the oak wood surfaces.

**Figure 4 nanomaterials-09-01568-f004:**
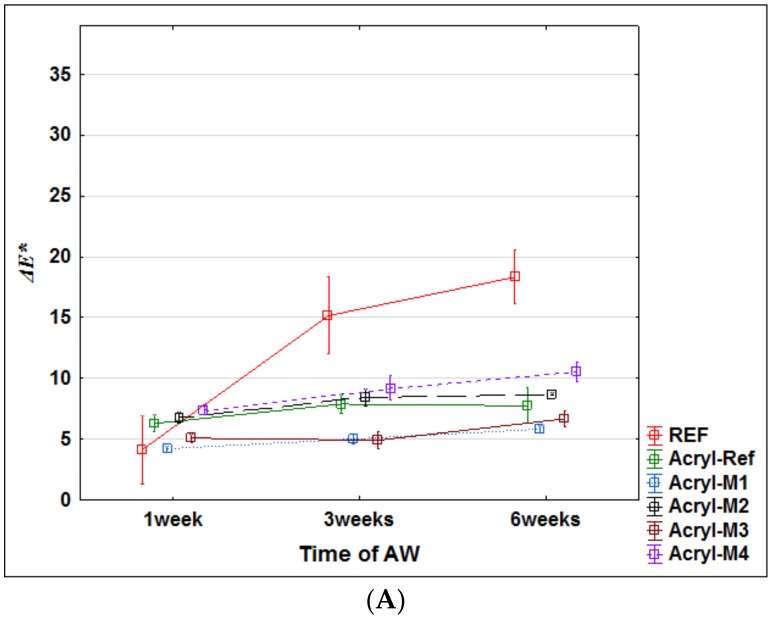
Total color change *ΔE** of the tested coatings during 6 weeks of weathering. (**A**—Acrylic coating; **B**—Oil-based coating; **C**—Synthetic thin layer coating).

**Figure 5 nanomaterials-09-01568-f005:**
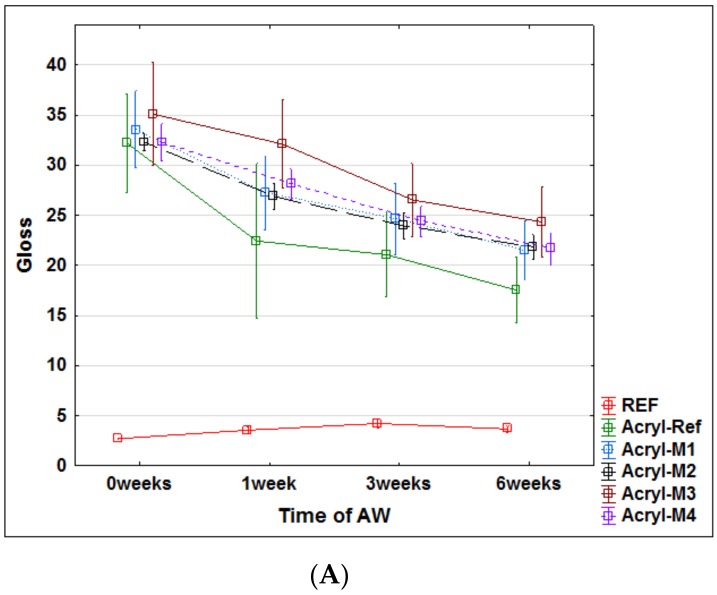
Gloss change of the tested coatings during 6 weeks of weathering. (**A**—Acrylic coating; **B**—Oil-based coating; **C**—Synthetic thin layer coating).

**Figure 6 nanomaterials-09-01568-f006:**
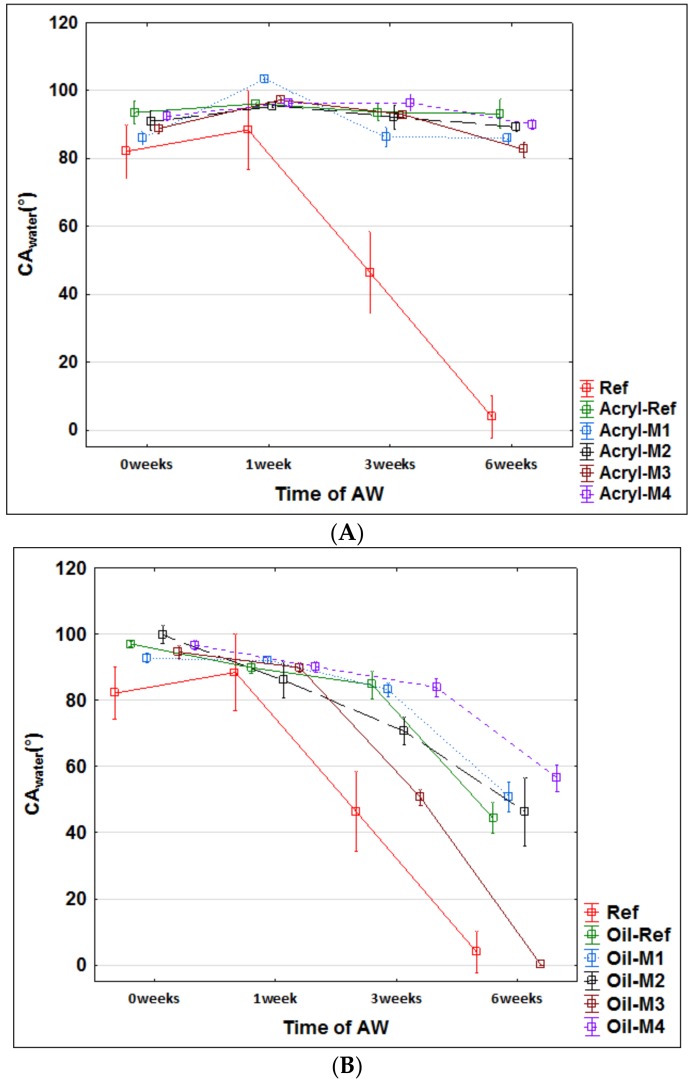
Water contact angle (CA_water_) change of the tested coatings during 6 weeks of weathering. (**A**—Acrylic coating; **B**—Oil-based coating; **C**—Synthetic thin layer coating).

**Figure 7 nanomaterials-09-01568-f007:**
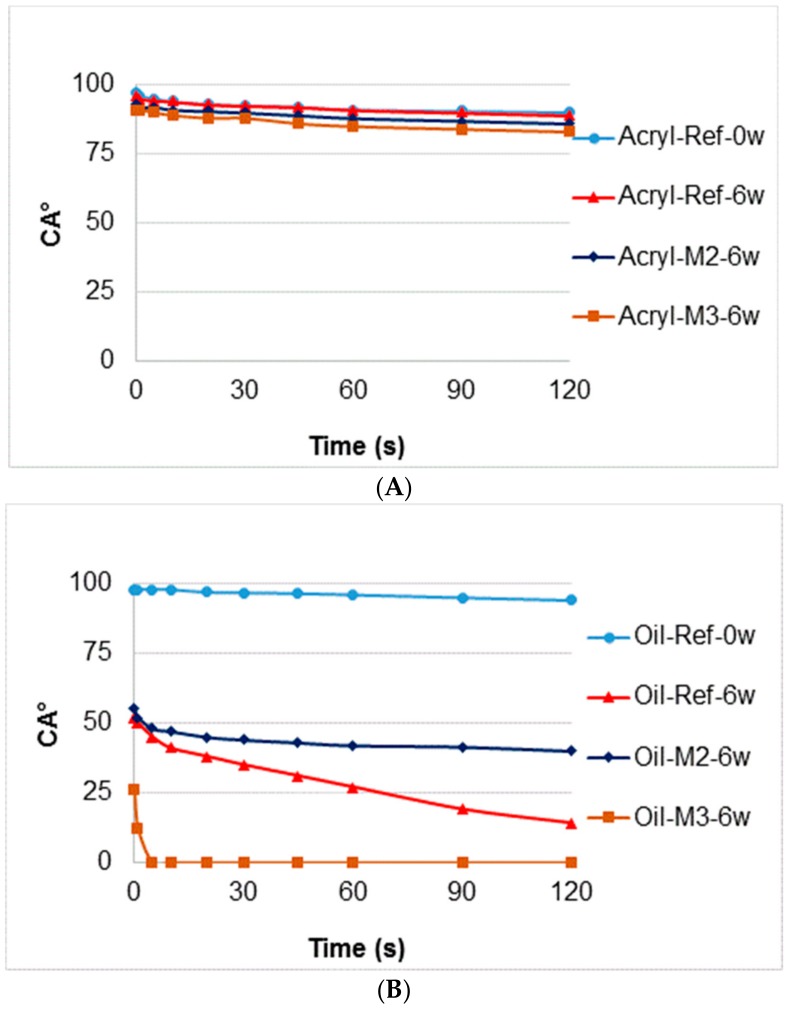
Dynamic water contact angle change of selected tested coatings before and after 6 weeks of weathering. (**A**—Acrylic coating; **B**—Oil-based coating; **C**—Synthetic thin layer coating).

**Figure 8 nanomaterials-09-01568-f008:**
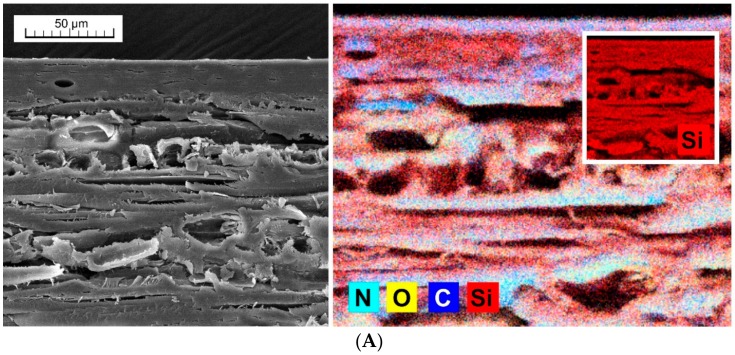
SEM and elemental analysis of selected tested coatings after accelerated artificial weathering. (**A**) Acrylic coating on unmodified oak wood (Acryl-Ref); (**B**) Acrylic coating on M2 modified oak wood (Acryl-M2); (**C**) synthetic thin layer coating on M3 modified oak wood (Synth-M3).

**Figure 9 nanomaterials-09-01568-f009:**
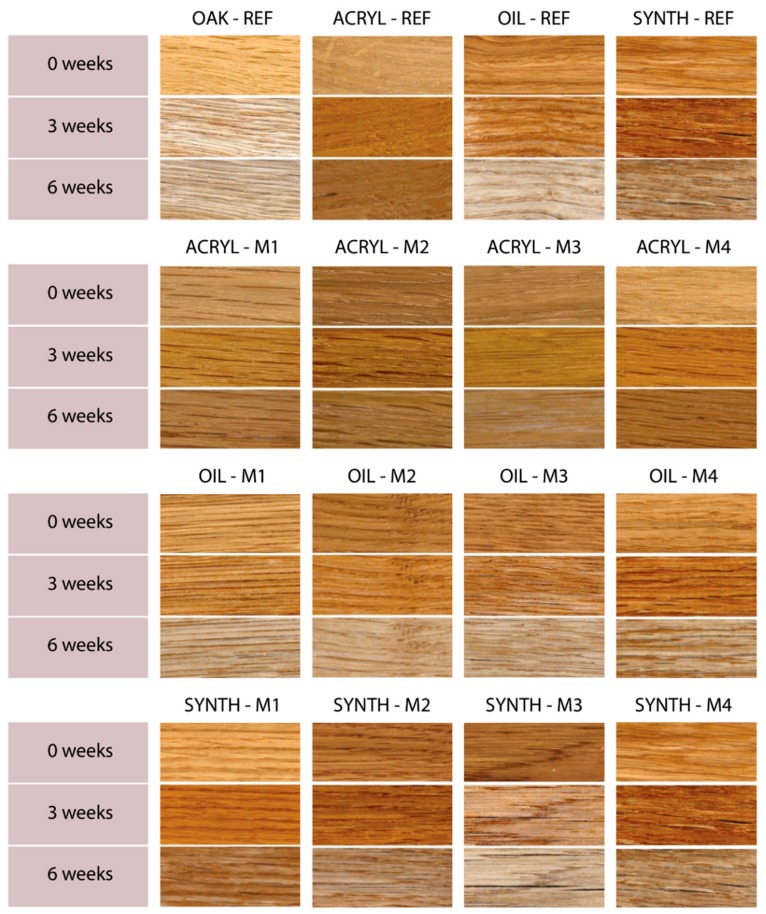
Visual evaluation of the tested coatings applied to surface modification (M1–M4) compared to unmodified oak wood (REF) during AW.

**Table 1 nanomaterials-09-01568-t001:** Specification of oak wood samples surface modifications.

Type of Surface Modification	Chemical Composition
Ref	Native Oak without modification
M1	UV light-stabilizer (commercial product): 2-(2-hydroxyphenyl)-benzotriazoles with HALS (bis(1,2,2,6,6-pentamethyl-4-piperidyl) sebacate & methyl 1,2,2,6,6-pentamethyl-4-piperidyl sebacate); all in 3% water solution
M2	UV light-stabilizer 2-(2-hydroxyphenyl)-benzotriazoles with HALS (M1) and nanoparticles of ZnO (25 nm)—weight ratio of UV stabilizers and nanoparticles in composition 1:1; all in 3% water dispersion concentration
M3	UV light-stabilizer 2-(2-hydroxyphenyl)-benzotriazoles with HALS (M1) and mixture of ZnO:TiO_2_ nanoparticles (in a 1:1 weight ratio); ZnO (25 nm) and TIO_2_ (6 nm) nanoparticles—weight ratio of UV stabilizers and nanoparticles in composition 1:1; all in 3% water dispersion concentration
M4	UV light-stabilizing penetration layer (commercial product) based on synthetic resins, organic UV light stabilizers, and IPBC fungicide

**Table 2 nanomaterials-09-01568-t002:** Specifications of the tested coatings.

Type of Coating	Specification of Composition
REF	native oak wood without modification and without coating system
Acryl	Acrylic thick layer exterior coating: Acrylate thick-layer water-solved glaze with fungicides (5-chlor-2-methylisothiazol-3(2H)-on) and UV-stabilizers
Oil	Oil-based film forming exterior coating: Transparent oil-based coating containing dis-aromatized white spirit, natural vegetable oils, 3-iodo-2-propynyl N-butylcarbamate (IPBC) as fungicide, UV-stabilizers
Synth	Synthetic thin layer exterior coating: mixture of synthetic resins and oils in organic solvents with additives (BIT as fungicide 0.5%) and butanonoxime (0.5%)

**Table 3 nanomaterials-09-01568-t003:** Mean values of initial color coordinates of the tested coatings systems (*n* = 12).

Color Coordinate	REF	Acryl					Oil					Synth				
		Ref	M1	M2	M3	M4	Ref	M1	M2	M3	M4	Ref	M1	M2	M3	M4
*L**	66.4	60.1	58.2	52.0	55.7	64.7	59.0	58.2	50.2	50.4	56.7	49.1	62.0	50.3	47.7	56.8
*a**	7.2	7.8	7.1	8.6	8.5	7.7	10.8	9.1	11.9	12.8	9.9	11.4	10.1	12.8	13.6	11.8
*b**	19.7	21.4	20.5	21.1	23.0	23.9	26.5	25.9	25.9	27.6	26.3	24.1	28.4	25.1	26.6	28.6

*L**, *a**, *b** are color coordinates (see part Section 2.4 of Materials and methods).

**Table 4 nanomaterials-09-01568-t004:** One cycle of weathering in a UV-chamber according to modified EN 927-6.

Weathering in UV-Chamber: One Cycle = 1 Week (168 h)			Functions
1st step	24 h		Temperature 45 ± 3 °C, Water-Spray (off), UV (off)
2nd step	A	2.5 h	Temperature 65 ± 3 °C, Water-Spray (off), UV Irradiance 1.10 W·m^−2^ at 340 nm
	B	0.5 h	Temperature 20 ± 1 °C, Water-Spray (on), UV (off)
	A + B	3 h	
	Sub-cycle (A + B): 48 sub-cycles × 3-h of one, i.e., together 144 h		

In a comparison according to EN 927-6: 2006, the UV-chamber parameters in the 2nd step/A are as follows: Temperature = 60 ± 3 °C, UV Irradiance = 0.89 W·m^−2^ at 340 nm.

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
