# Peer review of "Durability of the Exterior Transparent Coatings on Nano-Photostabilized English Oak Wood and Possibility of Its Prediction before Artificial Accelerated Weathering"

_nanomaterials, 2019, doi:10.3390/nano9111568_

Round 1

Reviewer 1 Report

The paper presents quite interesting results, referring to the study of the durability of transparent exterior coatings, by using the surface modification of wood with UV-stabilizing substances.

However, I think that the introduction need to be improved , since it lacks of a deep study on the recent literature on the same topic. I suggest the authors to deeply study the present literature about the development and characterization of nanostructured innovative photo-polymerizable coatings for different substrate (wood, stone , glass). I suggest the authors to read recent international paper about this topic and to add them to the references. In particular, there is one interesting  paper about this topic, published on POC , i.e. Progress in Organic Coatings,
Volume 99, 2016, Pages 230-239. In addition, I would like to suggest to the authors apply the innovative coatings to a selected wood substrate in order to verity the hydrophobic properties, by measuring dynamic water contact angle and water capillarity properties, as a function of time exposure to outdoor conditions . I suggest to evaluate the colorimetric variation of the treated substrate after the application of the coatings and the penetration depth of the coatings inside the substrate by SEM analysis.

Finally, referring to Line 119: "Two samples were tested for each type of surface modification and coating system", I think that the number of the tested samples is too low to have a significant statistical analysis.  Please increase the number of analysed samples .

I also suggest the authors to improve the English form.

I finally recommend the publication of the paper after these minor revisions.

Author Response

The authors added the comments and answers for reviewer in Word document.

Reviewer 2 Report

The paper is written at a fairly high scientific level. The results are described in detail and comments are given.

It is desirable to describe in more detail the properties of the titania particles, since the crystalline modification is important for the photocatalyst. The authors used very complex compositions for wood impregnation, so it is difficult to identify the effect of each component.
The main contribution to the contact angle is made by the type of coating (acrylic, oil). Therefore, it is difficult to separate the impregnation effect directly. Moreover, the porous and rough surface introduce a large error in the determination of the contact angle.
From the materials of the manuscript, it is difficult to determine the thickness of each type of coating. The authors do not give the composition of HALS.

Author Response

(The authors gave the same response as above.)
